# Foreground Confusion under Domain Shift: The Hidden Bottleneck in Source-Free Domain Adaptive Object Detection

## Abstract

Source-Free Domain Adaptive Object Detection (SFOD) adapts detectors to new domains without source data, which is vital when privacy or storage constraints apply. SFOD is hindered by two key challenges: unreliable pseudo-labels, and *foreground-background confusion*, which occurs when domain shift induces spurious background activations that degrade localization and, in turn, classification. We introduce **FOCUS-SFOD**, a lightweight, architecture-agnostic framework with two complementary losses: **CLEAN** (Consistency Loss for Eliminating Activation Noise) mitigates foreground-background confusion by aligning channel-mean maps with simple foreground priors, improving localization; **PAERL** (Peak-Adjusted Entropy-Regularized Loss) reduces sensitivity to noisy pseudo class-labels by down-weighting trivial teacher-student agreements, encouraging learning on harder or underrepresented categories. To the best of our knowledge, we are the first to formalize foreground-background confusion in SFOD and provide a risk-bound analysis linking CLEAN and PAERL to tighter localization and classification errors. Across strong baselines and diverse shifts, FOCUS-SFOD delivers consistent gains of up to +3.9 mAP, with zero inference overhead.

## 1 Introduction

Object detection drives critical computer vision applications in robotics, autonomous driving, and aerial imagery. Deep detectors (Ren et al., 2016; Liu et al., 2016; Redmon et al., 2016; Carion et al., 2020; Zhu et al., 2020) and large-scale datasets (Lin et al., 2014; Shao et al., 2019; Xia et al., 2018) have driven rapid progress in object detection. Yet, the performance of existing detectors degrades under domain shifts (Oza et al., 2023; Liu et al., 2024a), including large Vision Langauge Models (VLMs) (Chhipa et al., 2024) that are not suitable for deployment in real-time, latency-sensitive applications. To address such shifts, Source-Free Domain Adaptive Object Detection (SFOD) (Vibashan et al., 2023; Liu et al., 2023a; Hao et al., 2024) adapts a pre-trained source detector to an unlabeled target domain without any access to the source dataset. This makes it crucial for scenarios where privacy, or storage constraints preclude access to source data.

SFOD remains hindered by two critical challenges. The *first and widely recognized* issue is the unreliability of pseudo-labels: due to the lack of supervision, detectors rely on self-generated pseudo-labels that are often noisy. Misassigned pseudo-labels propagate errors during training, corrupting both classification and localization. Considerable effort has thus been devoted to pseudo-label refinement (Li et al., 2021a;b; 2022; Vibashan et al., 2023; Liu et al., 2023a; Hao et al., 2024).

However, beyond pseudo-label noise, a more fundamental yet overlooked bottleneck exists: *domain shift causes the feature space to become entangled between foreground and background regions*. This manifests as erroneous spatial activations on irrelevant background clutter (See Fig. 1), leading to: (i) *localization degradation* (bounding boxes misaligned or spread across irrelevant regions), (ii) *false positives* triggered by background textures, and (iii) *missed detections* when true objects are masked or suppressed by clutter (Appendix A.1 provides more examples). Mislocalized features are then fed into the classifier, exacerbating misclassification. While prior SFOD works overwhelmingly attribute this failure to pseudo-label noise (Vibashan et al., 2023; Zhang et al., 2023; Hao et al., 2024),

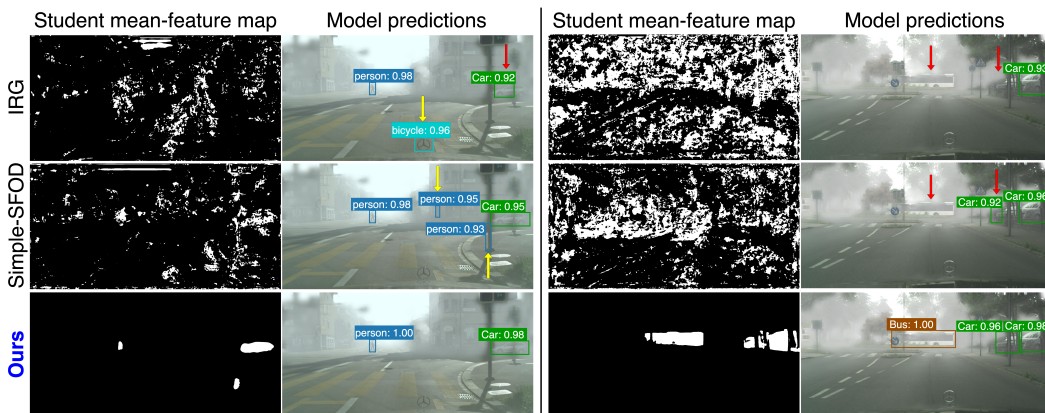

Figure 1: **Foreground-background confusion in SFOD and our remedy.** Two examples from the Foggy Cityscapes (Sakaridis et al., 2018) target set. In each example, the two columns show the student's channel-mean feature map from the last backbone layer (brighter = higher activation) and the model predictions, respectively. The state-of-the-art baselines IRG (Vibashan et al., 2023) and Simple-SFOD (Hao et al., 2024) exhibit strong background activations, leading to missed detections or localization errors (red arrows) and false positives (yellow arrows). Our method yields compact, object-shaped activations with tighter boxes and correct labels. *Zoom in for best view.*

we emphasize that *foreground-background confusion under domain shift is the more fundamental obstacle* that must be addressed to enable reliable SFOD.

To address these issues, we propose *FOCUS-SFOD* (FOreground-foCus and Unreliable-label Suppression), which integrates two complementary components designed to tackle the foreground-background confusion and the noisy pseudo class-labels. **CLEAN** (Consistency Loss for Eliminating Activation Noise) mitigates foreground-background confusion by aligning the model's channel-mean activation maps with simple foreground priors. This directly cleans feature-space activations and improves localization. **PAERL** (Peak-Adjusted Entropy-Regularized Loss) improves robustness to noisy pseudo class-labels by mitigating confirmation bias (down-weighting trivial teacher-student agreements), encouraging learning on harder or underrepresented categories, and employing a mild entropy regularizer to avoid head-class dominance. We also provide a theoretical risk-bound analysis, formally linking PAERL and CLEAN to tighter classification and localization error bounds. Together, these form a lightweight, theoretically bounded, architecture-agnostic framework that improves detection quality under domain shift. Our key contributions are:

1. To the best of our knowledge, we are the first to identify and formalize *foreground-background confusion* in SFOD, showing its central role in degraded localization and overall detection.

2. To address the above issue, we propose the use of CLEAN, a mask-agnostic regularizer that reduces spurious background activations, producing clean activations and better localization. We also propose PAERL, a new loss that mitigates pseudo class-label noise by reducing confirmation bias and encouraging learning on difficult/rare categories.

3. We provide the first theoretical risk-bound analysis for SFOD (to the best of our knowledge), formally linking PAERL and CLEAN to tighter classification and localization error bounds.

4. We comprehensively validate our framework through extensive experiments across strong baselines and diverse domain shifts, achieving between $+1.6$ to $+3.9$ mAP across datasets, with minimal computational overhead.

## 2 RELATED WORK

**Unsupervised Domain-Adaptive Object Detection (UDAOD).** UDAOD adapts a source-trained detector to an unlabeled target domain when source data is available during adaptation (Khodabandeh et al., 2019; Vs et al., 2021; Munir et al., 2021; Kennerley et al., 2024; Li et al., 2025). Recent works like DINO Teacher (Lavoie et al., 2025) leverage foundation models by training a source-domain

labeller with a frozen DINOv2 backbone and then aligning the student's patch features to a frozen DINO encoder. While this approach enriches supervision, it remains computationally demanding and does not address the issue of *foreground-background confusion*. Our CLEAN directly targets this problem through a simpler mechanism: we precompute binary foreground masks for the target dataset once and align the student's channel-mean activations to them. This one-time preprocessing makes CLEAN computationally efficient, budget-flexible (using either internal activation maps or external priors), and directly focused on improving localization under domain shift.

**Source-Free Domain-Adaptive Object Detection (SFOD).** SFOD eliminates the need to access source data during adaptation (Vibashan et al., 2023; Liu et al., 2023a; Hao et al., 2024). IRG (Vibashan et al., 2023) refines pseudo-labels via instance-relation graphs, PETS (Liu et al., 2023a) stabilizes teacher-student training with periodic exchanges, and *Simple-SFOD* (Hao et al., 2024) shows that careful self-training design can outperform complex architectures. While these works focus on improving the pseudo-label quality, they overlook the fundamental issue of foreground-background confusion in SFOD. Our proposed model, FOCUS-SFOD, addresses both the problems in a lightweight (simple-by-design, zero inference overhead), effective manner.

**Noise-Robust Learning with Noisy Labels.** Noisy-label learning introduces objectives that suppress the impact of corrupted labels (Wang et al., 2024). However, a direct adaptation of (Wang et al., 2024) fails in SFOD (See Appendix A.7) due to proposal imbalance, heavy class skew, and confirmation bias in teacher-student agreements. We extend it with foreground/background weighting, and entropy regularization, which together make PAERL effective for detection (See Table. 9). In combination, PAERL reduces pseudo class-label noise and CLEAN mitigates foreground-background confusion, yielding a lightweight, complementary framework for robust source-free adaptation.

## 3 METHODOLOGY

### 3.1 PRELIMINARIES

**Problem Statement.** We denote the labeled source-domain dataset as $\mathcal{D}_S = \{(x_i^s, \mathcal{Y}_i^s)\}_{i=1}^{N_S}$, where $x_i^s$ denotes the $i^{\text{th}}$ source image and $\mathcal{Y}_i^s$ is the corresponding ground-truth annotation containing bounding-box locations and class labels. $\mathcal{Y}_i^s = \{(b_{ij}, c_{ij})\}_{j=1}^{O_i}$, where $b_{ij} \in \mathbb{R}^4$ denotes bounding-box coordinates, $c_{ij} \in \{0, \ldots, K\}$ denotes the class label for the $j^{\text{th}}$ object, and $O_i$ denotes the number of objects in image $x_i^s$. The unlabeled target-domain dataset is denoted by $\mathcal{D}_T = \{x_i^t\}_{i=1}^{N_T}$. The cardinalities of source and target domain images are denoted by $N_S$ and $N_T$ respectively. An object detector can be expressed as $h(x) = f(g(x))$, where $g$ is the feature extractor and $f = (f_c, f_r)$ contains the: (i) *Classification head*: $f_c(g(x)) \in \Delta^K$ (a softmax over $K+1$ classes, with the extra class for background); and (ii) *Regression head*: $f_r(g(x)) \in \mathbb{R}^4$, predicting the bounding-box coordinates. The detector is usually trained by minimizing the combination of classification and regression loss terms: $\mathcal{L} = \mathcal{L}_{\text{cls}} + \mathcal{L}_{\text{reg}}$. Most existing efforts employ cross-entropy (or a variant) for the classification term and an $L_1$-style loss for the regression term. The goal of **SFOD** is to adapt a detector $h_{pre}$ trained on labeled source data $\mathcal{D}_S$ to unlabeled target data $\mathcal{D}_T$, without any further access to $\mathcal{D}_S$.

**'Mean Teacher'-based Self-Training Framework.** Most existing SFOD algorithms, including the state-of-the-art, adopt a two-stream *teacher–student* strategy that follows the 'Mean-Teacher (MT)' paradigm. A *student* detector $h^{st}$ with parameters $\mathbf{\Theta}^{st}$ is updated via gradient descent, and a *teacher* detector $h^{te}$ with parameters $\mathbf{\Theta}^{te}$ tracks the student through an exponential moving average (EMA). For every unlabeled target image $x_i^t$, a weak augmentation $\tilde{x}_i^t$ is first applied and then passed through the teacher to obtain a set of region proposals. After standard post-processing operations such as score filtering, non-maximum suppression (NMS), and a confidence threshold, the remaining proposals constitute the pseudo-annotation $\hat{\mathcal{Y}}_i^t = \{(\hat{b}_{ij}, \hat{c}_{ij})\}_{j=1}^{\hat{O}_i}$, where $\hat{b}_{ij}$, $\hat{c}_{ij}$ are pseudo bounding-boxes and corresponding pseudo class-labels respectively. $\hat{O}_i$ is the number of pseudo-annotations on the $i^{th}$ image. The student is then trained on a *strongly* augmented view $\bar{x}_i^t$ by minimizing the sum of RPN and RoI losses w.r.t. these pseudo labels:

$$\mathcal{L}_{\text{MT}} = \mathcal{L}_{\text{rpn}}(\bar{x}_i^t, \hat{\mathcal{Y}}_i^t) + \mathcal{L}_{\text{roi}}(\bar{x}_i^t, \hat{\mathcal{Y}}_i^t). \tag{1}$$

The network parameters are updated by:

$$\mathbf{\Theta}^{st} \leftarrow \mathbf{\Theta}^{st} - \eta \nabla_{\mathbf{\Theta}^{st}} \mathcal{L}_{\text{MT}}, \qquad \mathbf{\Theta}^{te} \leftarrow \lambda \mathbf{\Theta}^{te} + (1 - \lambda) \mathbf{\Theta}^{st}. \tag{2}$$

where $\eta$ is the learning rate and $\lambda \in (0, 1)$ is the EMA decay factor.

**Theory-guided objective (forward reference to Sec. 4).** The detection-risk decomposition in Sec. 4 will show that training on noisy pseudo-labels inflates (i) the *classification* risk by a factor $1/\lambda$ (Lemma 1, Theorem 1) and (ii) the *localization* risk through two additive terms, the deviation $\eta_{\text{reg}}$ and the miss-rate $\zeta$ (Lemma 2, Theorem 1). We therefore design two complementary modules: a peak-adjusted classification loss (**PAERL**) that replaces the multiplicative inflation with a tighter additive term (Theorem 2), and a spatial-focus regularizer (**CLEAN**) that directly shrinks $\eta_{\text{reg}}$ and $\zeta$ by mitigating foreground-background confusion.

## 3.2 FOCUS-SFOD

**Motivation and Overview.** As observed in (Vibashan et al., 2023; Liu et al., 2023a; Hao et al., 2024), a prevalent problem in mean-teacher self-training is that pseudo class-labels tend to be noisy due to domain shifts. *Beyond this well-known issue, we are the first to identify and formalize that domain shift induces a **foreground–background confusion** in SFOD: spatial activations for true objects are diluted by background clutter, which degrades localization and cascades into misclassification* (see Fig. 1). To jointly address unreliable pseudo-labels and foreground-background confusion, we propose **FOCUS-SFOD** (FOreground-foCus and Unreliable pseudo-label Supression), coupling two dedicated objectives within the standard Mean-Teacher framework: **PAERL** for pseudo class-label robustness and **CLEAN** for spatial focus regularization. FOCUS-SFOD integrates these losses with the conventional localization objective to enhance robustness during source-free adaptation.

**Peak-Adjusted Entropy-Regularised Loss (PAERL).** Most SFOD methods minimize cross-entropy loss (Vibashan et al., 2023; Liu et al., 2023a; Hao et al., 2024), which has an unbounded gradient that allows a single corrupted pseudo class-label to dominate training. We therefore introduce PAERL, which mitigates this negative impact of noisy pseudo class-labels by adaptively recalibrating the per-box classification loss. For student probabilities $f_c^{st} \in \mathbb{R}^{K+1}$, let $\mathbf{p} = f_c^{st}$ and $t = \arg\max_k p_k$. The following transform rescales the peak while making sure that elements of $\mathbf{p}'$ sum up to 1.

$$p'_k = \begin{cases} \dfrac{p_k + m}{1 + m}, & k = t, \\[2mm] \dfrac{p_k}{1 + m}, & k \neq t, \end{cases} \tag{3}$$

where $m$ is a large real value.

For each image $x_i^t$, we minimize

$$\mathcal{L}_{\text{PAERL}} = \Big[ \sum_{(\hat{b},\hat{c}) \in \hat{\mathcal{Y}}_i^t} w_{\hat{c}}(\alpha\,(-\log p'_{\hat{c}}) + \beta\,(1 - p_{\hat{c}})) \Big] + \gamma\, D_{\text{KL}}(\bar{\mathbf{p}} \,\|\, \mathcal{U}_{\mathcal{K}}). \tag{4}$$

where

$$\mathcal{K} := \{0, \ldots, K-1\}; \qquad Z := \sum_{(\hat{b},\hat{c}) \in \hat{\mathcal{Y}}_i^t} \sum_{k \in \mathcal{K}} p_k;$$

$$\bar{p}_k := \frac{1}{Z} \sum_{(\hat{b},\hat{c}) \in \hat{\mathcal{Y}}_i^t} p_k, \quad \forall k \in \mathcal{K}; \qquad D_{\text{KL}}(\bar{\mathbf{p}} \,\|\, \mathcal{U}_{\mathcal{K}}) = \log |\mathcal{K}| + \sum_{k \in \mathcal{K}} \bar{p}_k \log \bar{p}_k.$$

**Why PAERL is intrinsically robust to noisy pseudo-labels?** The *peak–adjust* operation in Eq. 3 moderates the student's logits by adding a large margin $m$ to its highest probability and then renormalising. This creates two mutually exclusive regimes: 1) **Teacher and student agree** ($\hat{c} = t$). The margin sits on the *same* logit that the loss differentiates, so the cross-entropy gradient for that box is uniformly scaled by the factor $p_{\hat{c}}/(p_{\hat{c}} + m) \ll 1$. Easy, likely-clean boxes therefore contribute vanishing updates, acting as a built-in *soft early-stopping* mechanism that prevents over-fitting to already-correct labels. 2) **Teacher and student disagree** ($\hat{c} \neq t$). The margin affects a *different* logit; the derivative with respect to the true class is unchanged, and the gradient reduces to the standard cross-entropy form. Hard or potentially mislabeled boxes thus retain a full corrective signal, allowing the student to challenge erroneous teacher guidance.

**Consistency Loss for Eliminating Activation Noise (CLEAN).** Domain shift entangles foreground and background in the spatial activations of the detector, producing foreground–background confusion that worsens localization and, by propagation, classification. To formalize and address this issue,

we introduce **CLEAN**. Consider the detector $h(x) = f(g(x))$ introduced in Sec. 3.1. The feature extractor outputs an activation tensor $a = g(x) \in \mathbb{R}^{H \times W \times C}$, where $H$, $W$, and $C$ are its height, width, and channel count. Taking the channel-wise mean yields $A \in \mathbb{R}^{H \times W}$, which highlights spatial locations with high average activations and thus usually traces foreground objects. Under a domain shift, however, this map becomes contaminated by background clutter: true objects may fade while background areas are spuriously accentuated (as illustrated in Fig. 1). We address this by aligning the student's mean map $A_S$ with a binary foreground prior $A_G$.

For each target image $x_i^t$ the prior yields a binary mask $A_G(x_i^t) \in [0,1]^{H' \times W'}$, and the student provides its channel-mean map $A_S(x_i^t) \in [0,1]^{H \times W}$. The student map is rescaled to match the dimensions of $A_G$ and their agreement is enforced with a *mean $\ell_1$* term and a Dice loss:

$$\mathcal{L}_{\text{CLEAN}}(x_i^t) = \frac{\lambda_1}{H' W'} \sum_{j,k} |A_{\text{S}}[j,k] - A_{\text{G}}[j,k]|$$

$$+ \ \lambda_2 \left( 1 - \frac{2 \sum_{j,k} A_{\text{S}}[j,k] \, A_{\text{G}}[j,k]}{\sum_{j,k} A_{\text{S}}[j,k] + \sum_{j,k} A_{\text{G}}[j,k] + \varepsilon} \right). \tag{5}$$

where $(j,k)$ index spatial positions, and $\varepsilon$ is a small constant for numerical stability.

**CLEAN is simple-by-design and budget-flexible.** CLEAN is intentionally minimal, operating only on the channel-mean activation map and a binary foreground prior, so it is architecture-agnostic and adds *no* inference-time cost. In practice:

- **Mask-agnostic prior.** $A_G$ can come from *any* class-agnostic foreground prior (Ren et al., 2024; Yuan et al., 2024; Lee et al., 2025) computed once per image before target adaptation. $A_G$ can also be obtained *internally* from the detector itself: thresholded channel-mean activations of the source-trained model.
- **One-time preprocessing.** Priors are precomputed and cached off-line, which adds a minimal compute overhead. See Appendix A.8 for more details on compute and memory costs.
- **Drop-in.** CLEAN plugs into any mean-teacher SFOD recipe and requires no additional heads, proposals, or architectural changes.

**Overall Objective.** The student network is trained to minimize

$$\mathcal{L} = \mathcal{L}_{\text{PAERL}} + \mathcal{L}_{\text{CLEAN}} + \mathcal{L}_{reg},$$

where $\mathcal{L}_{reg}$ denotes the detector's standard localization loss. $\mathcal{L}_{\text{PAERL}}$ addresses pseudo class-label noise by adjusting the student's confidence, blending cross-entropy with MAE, and applying a foreground/background weighting, whereas $\mathcal{L}_{\text{CLEAN}}$ tackles *foreground-background confusion* by aligning the channel-mean activation map with a foreground prior. Acting on complementary axes, label confidence and spatial focus, these losses *follow the factors in the detection bound* (Sec. 4): PAERL tightens the classification term (Theorem 2), while CLEAN shrinks the localization addends $\eta_{\text{reg}}$ and $2\zeta$ (Lemma 2, Theorem 1). See Appendix A.3 for FOCUS-SFOD pseudo-code.

## 4 THEORETICAL INSIGHTS

This section formalizes how the modules introduced in Sec. 3 target specific terms in the detection risk. We first decompose risk under teacher-generated pseudo-labels, then show that a peak-adjusted classification objective yields a tighter classification term and explain how spatial confusion appears additively in the localization term, precisely what CLEAN is designed to reduce.

For a given dataset $\mathcal{D} = \{(x_i, y_i)\}_{i=1}^N$, we define an object set for image $i$ as $\mathcal{Y}_i = \{(b_{ij}, c_{ij})\}_{j=1}^{O_i}$, the pseudo annotations as $\hat{\mathcal{Y}}_i = \{(\hat{b_{ij}}, \hat{c_{ij}})\}_{j=1}^{\hat{O}_i}$ and the detection risk is shown in Eq. 6. We implicitly index over objects inside each image. We write $f(g(x))$ simply as $f(x)$ for brevity, $f_c \colon x \mapsto (p_0(x), \ldots, p_K(x))$, and $\mathcal{D}_{X,C}$ and $\mathcal{D}_{X,B}$ as the marginal of $\mathcal{D}$ over the image-class $(x,c)$ pairs and image-box pairs $(x,b)$, respectively.

$$R_{\mathcal{D}}^{det}(f) = R_{\mathcal{D},clean}^{cls}(f_c) + R_{\mathcal{D},clean}^{reg}(f_r) = \mathbb{E}_{(x,c) \sim \mathcal{D}_{X,C}}[-\log p_c(x)] + \mathbb{E}_{(x,b) \sim \mathcal{D}_{X,B}} \lVert f_r(x) - b \rVert_1 \tag{6}$$

where $R_{\mathcal{D},clean}^{cls}$ is the standard cross entropy loss and $R_{\mathcal{D},clean}^{reg}$ is the standard $L_1$ regression loss used in most of the object detectors. We now state in Lemma 1 the classification risk under noisy pseudo class-labels.

**Lemma 1.** *Let $\mathcal{D}^T$ be the target distribution over $(x, c)$, and let the pseudo-label $\hat{c}$ be drawn from an* arbitrary *class-conditional transition matrix $T \in [0, 1]^{(K+1) \times (K+1)}$ with $\sum_{i=0}^{K} T_{ji} = 1$ for every $j$ and $\lambda = \min_j T_{jj} > 0$. For any classifier $f_c^{st} : x \mapsto p(x) = (p_0(x), \ldots, p_K(x))$ and $R_{\mathcal{D}^T, clean}^{cls}(f_c^{st}) = \mathbb{E}_{(x,c) \sim \mathcal{D}^T}[-\log p_c(x)]$, $R_{\mathcal{D}^T, noisy}^{cls}(f_c^{st}) = \mathbb{E}_{(x,\hat{c})}[-\log p_{\hat{c}}(x)]$, we have the following relationship*

$$R_{\mathcal{D}^T, clean}^{cls}(f_c^{st}) \leq \frac{1}{\lambda} R_{\mathcal{D}^T, noisy}^{cls}(f_c^{st}). \tag{7}$$

All the proofs for the theoretical analysis are provided in the Appendix A.2. Lemma 1 employs the class-conditional transition matrix $T \in [0, 1]^{(K+1) \times (K+1)}$, where $T_{ji} = \Pr[\hat{c} = i \mid c = j]$ represents the label noise induced by the mean-teacher: an exponential-moving-average (EMA) copy of the student whose deterministic predictions form a stochastic channel over the data distribution (Tarvainen & Valpola, 2017). The diagonal element $T_{jj}$ is the teacher's per-class hit-rate, and we define $\lambda = \min_j T_{jj} > 0$ to rule out classes the teacher never recognizes, a minimal condition for identifiability of the clean risk under arbitrary label noise (Liu et al., 2023b). Lemma 1 therefore shows that mean-teacher asymmetry costs only a multiplicative factor $1/\lambda$; when the teacher is perfect ($\lambda = 1$) the bound reduces to the standard clean-risk expression. Next, we show in Lemma 2 the regression risk under noisy pseudo bounding box labels.

**Lemma 2.** *Let $\mathcal{D}^T$ be the target-domain distribution over image–box pairs $(x, b)$ and let $f_r^{te}$ be the teacher regressor that outputs a pseudo-box $\hat{b} = f_r^{te}(x)$. For every ground-truth box define the indicator $M(x, b) = \mathbb{1}[\text{IoU}(\hat{b}, b) \geq \tau] \in \{0, 1\}$, i.e. $M = 1$ when the teacher matches the ground truth box under the usual IoU threshold $\tau$, and $M = 0$ otherwise. Assume all boxes are normalized to the unit square, so that $\|u - v\|_1 \leq 2$ for any two boxes $u, v$. Define*

$$R_{\mathcal{D}^T, noisy}^{reg}(f_r^{st}) = \mathbb{E}_{(x,b)}[M \|f_r^{st}(x) - \hat{b}\|_1], \eta_{reg} = \mathbb{E}_{(x,b)}[M \|\hat{b} - b\|_1], \zeta = \mathbb{E}_{(x,b)}[1 - M].$$

*Then for any student regressor $f_r^{st}$*

$$R_{\mathcal{D}^T, clean}^{reg}(f_r^{st}) \leq R_{\mathcal{D}^T, noisy}^{reg}(f_r^{st}) + \eta_{reg} + 2\zeta. \tag{8}$$

Lemma 2 expresses the clean localization risk as the sum of the noisy risk and the single constant $\eta_{\text{reg}}$, defined as the teacher's expected $L_1$ deviation from ground truth and entirely determined by the geometry of boxes; no distributional assumptions are introduced. Because the argument is purely metric, the bound holds regardless of how pseudo-boxes are generated or how label noise correlates across objects and classes. Importantly for SFOD, domain-shift–induced *foreground–background confusion* increases both the teacher's miss-rate $\zeta$ and the deviation term $\eta_{\text{reg}}$ by spreading activations into cluttered background, thereby degrading localization (precisely what CLEAN is designed to mitigate in Eq. 5). We now leverage Lemma 1 and Lemma 2 to state the upper bound on detection risk as seen in Theorem 1.

**Theorem 1.** *Given pseudo class-labels generated by a teacher with transition matrix $T$ satisfying $\lambda = \min_j T_{jj} > 0$ and bounding-box pseudo-labels satisfying the noise rate $\eta_{reg}$ and let $\zeta = \mathbb{E}_{(x,b)}[1 - M(x, b)]$ be the teacher's miss-rate for ground-truth boxes. Then, for any student heads $(f_c^{st}, f_r^{st})$, we have*

$$R_{\mathcal{D}^T}^{det}(f_c^{st}, f_r^{st}) \leq \frac{1}{\lambda} R_{\mathcal{D}^T, noisy}^{cls}(f_c^{st}) + R_{\mathcal{D}^T, noisy}^{reg}(f_r^{st}) + \eta_{reg} + 2\zeta. \tag{9}$$

Theorem 1 adds the two sources of error. The classification part is inflated by $1/\lambda$, while the regression part is simply shifted by $\eta_{\text{reg}}$ plus the miss-rate penalty $2\zeta$. Foreground–background confusion acts precisely through these localization terms, motivating an explicit spatial regularizer that reduces $\eta_{\text{reg}}$ and $\zeta$ by cleaning activations (CLEAN, Eq. 5).

**Theorem 2.** *Let $f_\eta^* = \arg\min_{f_c^{st}} R_L^\eta(f_c^{st})$ be the population minimizer of the peak-adjusted classification loss $R_L^\eta$ under the teacher-noise model $T$. We define $R_L^\eta(f_c^{st}) = \mathbb{E}_{(x,c)}\Big[(1 - \eta_x) L(f_c^{st}(x), c) + \sum_{k \neq c} \eta_{x,k} L(f_c^{st}(x), k)\Big]$. where $L$ is any classification loss that satisfies $\left|\sum_{k=1}^{K}(L(u_1, k) - L(u_2, k))\right| \leq \delta$ whenever $\|u_1 - u_2\|_2 \leq \epsilon$, and $\delta \to 0$ as $\epsilon \to 0$,*

$\eta_{x,k} := \Pr_T(\hat{c} = k \mid x), \eta_c := \sum_{k \neq c} \eta_{x,k}.\ w = \mathbb{E}_x(1 - \eta_c), a = \min_{x,k}(1 - \eta_c - \eta_{x,k}),$, *and let* $\zeta = \mathbb{E}_{(x,b)}[1 - M(x,b)]$ *be the teacher's* miss-rate *for ground-truth boxes, as introduced in Lemma 2. Then, for any student regression head* $f_r^{st}$,

$$R_{\mathcal{D}^T}^{\det}(f_\eta^*, f_r^{st}) \leq \left(2\delta + \frac{2w\delta}{a}\right) + R_{\mathcal{D}^T, noisy}^{reg}(f_r^{st}) + \eta_{reg} + 2\zeta. \qquad (10)$$

Theorem 2 replaces the multiplicative factor $1/\lambda$ in Theorem 1 with the additive term $2\delta + \frac{2w\delta}{a}$, thus making it tighter. Because $\delta \to 0$ as $\varepsilon \to 0$, this additive bound becomes arbitrarily tight even for moderate $\lambda$; moreover, since $1/\lambda \geq 1$, it is *strictly tighter* than the original multiplicative bound whenever the teacher is imperfect ($\lambda < 1$).

## 5 EXPERIMENTS

**Datasets, Metrics and Baselines.** We integrate our method into three recent state-of-the-art SFOD methods: IRG (Vibashan et al., 2023), PETS (Liu et al., 2023a), and Simple-SFOD (Hao et al., 2024). Following existing works, we report mean average precision (mAP) with an IoU threshold of 0.5. We use five publicly available data sets - Cityscapes (Cordts et al., 2016), Foggy Cityscapes (Sakaridis et al., 2018), KITTI (Geiger et al., 2013), Sim10k (Johnson-Roberson et al., 2016), BDD100k (Yu et al., 2018) which cover four challenging real-world domain change scenarios. See Appendix A.5 for detailed description on datasets.

**Implementation Details.** For a fair comparison, we first reproduce each baseline following its published code and hyper-parameters and apply our components on top of these reproduced baselines, without altering any training configuration. Faster R-CNN is adopted as the base detector in all cases; IRG uses a ResNet-50 backbone, whereas PETS and Simple-SFOD uses VGG-16. The original batch sizes, epochs, learning rates, optimization schedules and data pre-processing pipelines are kept unchanged. Experiments for IRG are run on a single NVIDIA V100 GPU, while PETS and Simple-SFOD are trained and evaluated on a single NVIDIA RTX A6000 GPU. Tables 1 - 5 use Yuan et al. (2024) for getting the class-agnostic binary foreground masks in CLEAN. We provide results with different mask sources in Table 12 (in Appendix A.9).

Table 1: Performance comparison on *Cityscapes→ Foggy Cityscapes* (C→F), *Sim10k→ Cityscapes* (S→C), and *Kitti→ Cityscapes* (K→C). "rep" = reproduced results; "+ Ours" = our method.

| Category | Method | C→F | | | | | | | | | S→C | K→C |
| | | prsn | rider | car | truck | bus | train | mcycle | bicycle | **mAP** | **AP Car** | **AP Car** |
|---|---|---|---|---|---|---|---|---|---|---|---|---|
| **ZeroShot** | Grounding-DINO (Liu et al., 2024b) (ECCV'24) | 37.3 | 15.3 | 56.5 | 28.2 | 43.1 | 1.6 | 28.3 | 46.2 | 32.1 | 40.4 | 40.4 |
| **S** | Source Only | 29.3 | 34.1 | 35.8 | 15.4 | 26.0 | 9.09 | 22.4 | 29.7 | 25.2 | 32.0 | 33.9 |
| **UDAOD** | SSAL (Munir et al., 2021) (NeurIPS'21) | 45.1 | 47.4 | 59.4 | 24.5 | 50 | 25.7 | 26 | 38.7 | 39.6 | 51.8 | 45.6 |
| | PT (Chen et al., 2022) (ICML'22) | 40.2 | 48.8 | 59.7 | 30.7 | 51.8 | 30.6 | 35.4 | 44.5 | 42.7 | 55.1 | 60.2 |
| | MTM (Weng & Yuan, 2024) (AAAI'24) | 51 | 53.4 | 67.2 | 37.2 | 54.4 | 41.6 | 38.4 | 47.7 | 48.9 | 58.1 | - |
| | SEEN-DA (Li et al., 2025) (CVPR'25) | 58.5 | 64.5 | 71.7 | 42 | 61.2 | 54.8 | 47.1 | 59.9 | 57.5 | 66.8 | 67.1 |
| **SFOD** | SFOD (Li et al., 2021a) (AAAI'21) | 21.7 | 44.0 | 40.4 | 32.2 | 11.8 | 25.3 | 34.5 | 34.3 | 30.6 | 42.3 | 43.6 |
| | SFOD-Mosaic (Li et al., 2021b) (AAAI'21) | 25.5 | 44.5 | 40.7 | 33.2 | 22.2 | 28.4 | 34.1 | 39.0 | 33.5 | 42.9 | 44.6 |
| | LODS (Li et al., 2022) (CVPR'22) | 34.0 | 45.7 | 48.8 | 27.3 | 39.7 | 19.6 | 33.2 | 37.8 | 35.8 | - | 43.9 |
| | IRG (rep) (Vibashan et al., 2023) (CVPR'23) | 36.9 | 45.7 | 51.5 | 26.4 | 41.3 | 25.7 | 29.1 | 39.1 | 37.0 | 45.9 | 47.2 |
| | **+ (Ours)** | 37 | 45.9 | 51.7 | 30.2 | 44.7 | 30 | 32.9 | 40.6 | **39.0** | **49.1** | **49.8** |
| | PETS (rep) (Liu et al., 2023a) (ICCV'23) | 46.1 | 52.6 | 63.5 | 21.8 | 46.8 | 5.5 | 37.1 | 48.7 | 40.3 | 57.5 | 46.7 |
| | **+ (Ours)** | 46.2 | 52.9 | 63.2 | 24 | 49.1 | 10.4 | 40.5 | 48.6 | **41.9** | **59.1** | **48.9** |
| | Simple-SFOD (rep) (Hao et al., 2024) (ECCV'24) | 40.9 | 48 | 58.9 | 29.6 | 51.9 | 50.2 | 36.2 | 44.1 | 45.0 | 55.4 | 46.2 |
| | **+ (Ours)** | 41 | 48.3 | 58.7 | 33.6 | 54.8 | 54.3 | 38.6 | 46.2 | **46.9** | **58.8** | **50.1** |

**Adaptation to Adverse Weather.** To evaluate the adverse weather domain shifts, we perform adaptation from Cityscapes to Foggy Cityscapes. Table 1 demonstrates that our approach consistently enhances performance (by ∼2 mAP) over the recent state-of-the-art SFOD approaches.

**Synthetic to Real-world.** We use the Sim10k dataset as the source domain and the "car" category from Cityscapes as the target domain (Table 1 S→C). Integrated with Simple-SFOD and IRG, our proposed approach achieves notable improvements of 3.4 and 3.2 mAP, respectively. When combined with PETS, which already surpasses existing SFOD methods by a significant margin (>10 mAP), our method further boosts performance by 1.6 mAP.

**Cross-camera Adaptation.** Table 1 shows that in cross-camera adaptation scenarios (K→C), our proposed approach consistently enhances detection performance, achieving a substantial improvement of 3.9 mAP on Simple-SFOD and a mean mAP of 2.9 across the three recent SFOD methods.

Table 2: *Cityscapes→ BDD100k*. "rep" = reproduced results; "+ Ours" = our method.

| Category | Method | truck | car | rider | person | motor | bicycle | bus | **mAP** |
|---|---|---|---|---|---|---|---|---|---|
| **ZeroShot** | Grounding-DINO (Liu et al., 2024b) (ECCV'24) | 32.3 | 66.1 | 31.4 | 38.1 | 17.9 | 28.7 | 25.4 | 34.3 |
| **S** | Source Only | 9.9 | 51.5 | 17.8 | 28.7 | 7.5 | 10.8 | 7.6 | 19.1 |
| **UDAOD** | SWDA (Saito et al., 2019) (CVPR'19) | 15.2 | 45.7 | 29.5 | 30.2 | 17.1 | 21.2 | 18.4 | 25.3 |
| | CR-DA-Det (Wang et al., 2021) (TIP'21) | 19.5 | 46.3 | 31.3 | 31.4 | 17.3 | 23.8 | 18.9 | 26.9 |
| | MTM (Weng & Yuan, 2024) (AAAI'24) | 53.7 | 35.1 | 68.8 | 23.0 | 28.8 | 23.8 | 28.0 | 37.3 |
| **SFOD** | SFOD (Li et al., 2021a) (AAAI'21) | 20.4 | 48.8 | 32.4 | 31.0 | 15.0 | 24.3 | 21.3 | 27.6 |
| | SFOD-Mosaic (Li et al., 2021b) (AAAI'21) | 20.6 | 50.4 | 32.6 | 32.4 | 18.9 | 25.0 | 23.4 | 29.0 |
| | A$^2$SFOD (Chu et al., 2023) (AAAI'23) | 26.6 | 50.2 | 36.3 | 33.2 | 22.5 | 28.2 | 24.4 | 31.6 |
| | IRG (rep) (Vibashan et al., 2023) (CVPR'23) | 31.4 | 59.7 | 32.8 | 39.9 | 16.7 | 26.9 | 21.5 | 32.7 |
| | **+ (Ours)** | 31.7 | 59.5 | 33.2 | 39.9 | 20.9 | 31.6 | 28.3 | **35** |
| | PETS (rep) (Liu et al., 2023a) (ICCV'23) | 19.3 | 62.4 | 34.5 | 42.6 | 17.0 | 26.3 | 16.9 | 31.3 |
| | **+ (Ours)** | 19.9 | 61.9 | 34.7 | 42.7 | 21.3 | 30.5 | 20.9 | **33.1** |
| | Simple-SFOD (rep) (Hao et al., 2024) (ECCV'24) | 32 | 60 | 33.4 | 40.2 | 19.7 | 29.9 | 24.9 | 34.3 |
| | **+ (Ours)** | 32.6 | 59.8 | 34.0 | 40.0 | 25.7 | 35.7 | 30.5 | **36.9** |

**Small-scale to Large-scale.** We select Cityscapes as the source domain and BDD100k as the target domain to study this shift. Following Liu et al. (2023a), we focus on the seven categories shared with Cityscapes. From Table 2, we can see that our method achieves an average boost of 2.2 mAP when integrated into recent SFOD methods.

**Results on Extreme Shifts.** To stress-test our method under severe domain shifts, we evaluate on three challenging transfers (using IRG as the baseline): i) Realistic to artistic data (PascalVOC (Everingham et al., 2010) → Clipart (Inoue et al., 2018), ii) RGB to Thermal (FLIR (Teledyne FLIR LLC, 2019) visible → infrared), and iii) Thermal to RGB (FLIR Infrared to COCO (Lin et al., 2014)). Tables 3, 4, and 5 show that our method consistently improves detection accuracy (by ∼2 mAP) even under extreme domain shifts, underscoring its robustness.

Table 3: Pascal VOC → Clipart results.

| Method | Aero | Bicycle | Bird | Boat | Bottle | Bus | Car | Cat | Chair | Cow | Table | Dog | Horse | Bike | Person | Plant | Sheep | Sofa | Train | TV | mAP |
|---|---|---|---|---|---|---|---|---|---|---|---|---|---|---|---|---|---|---|---|---|---|
| IRG (rep) | 23.0 | 58.2 | 28.6 | 21.4 | 29.5 | 58.6 | 40.0 | 9.1 | 37.4 | 15.3 | 27.4 | 11.3 | 38.6 | 56.6 | 53.0 | 41.7 | 15.1 | 21.5 | 34.0 | 36.1 | 32.8 |
| IRG + Ours | 24.5 | 53.6 | 27.1 | 24.9 | 34.0 | 64.9 | 40.2 | 9.1 | 40.0 | 28.1 | 22.4 | 8.2 | 37.3 | 77.4 | 48.9 | 48.2 | 9.1 | 20.6 | 44.7 | 38.8 | **35.1** |

Table 4: FLIR Visible → Infrared results.

| Method | person | bicycle | car | **mAP** |
|---|---|---|---|---|
| IRG | 59.8 | 42.3 | 68.1 | 56.7 |
| IRG + Ours | 61.9 | 43.7 | 69.9 | **58.5** |

Table 5: FLIR Infrared → COCO results.

| Method | person | bicycle | car | **mAP** |
|---|---|---|---|---|
| IRG (re-run) | 25.2 | 12.7 | 20.3 | 19.4 |
| IRG + Ours | 27.1 | 13.8 | 21.9 | **20.9** |

**Effect of PAERL and CLEAN Losses - Class vs. Localization Gains.** Figure 2 depicts three metrics: standard mAP-50, Class-only AP-50 (predictions snapped to nearest GT boxes; isolates classification accuracy), and Box-only AP-50 (class-agnostic evaluation; isolates localization accuracy). As shown, PAERL substantially improves Class-only AP, confirming its effectiveness in reducing noisy pseudo-class labels, while CLEAN notably boosts Box-only AP, validating improved spatial alignment. The combined approach achieves the highest standard mAP-50, indicating complementary benefits from addressing both the issues. See Appendix A.6 for more details.

**Ablation Studies.** Table 6 demonstrates the impact of integrating CLEAN and PAERL into IRG across two domain shift scenarios - Cityscapes to Foggy Cityscapes and Sim10k to Cityscapes. We observe that individually adding CLEAN or PAERL consistently improves the baseline performance, achieving a +0.8 and +1.0 increase in mAP for Cityscapes to Foggy Cityscapes, and +1.6 and +1.5 AP for car detection from Sim10k to Cityscapes, respectively. Furthermore, combining these components yields the best results, enhancing mAP by +2.0 and AP by +3.2 in the respective domain shifts. These results confirm the complementary nature of CLEAN and PAERL when integrated into the baseline.

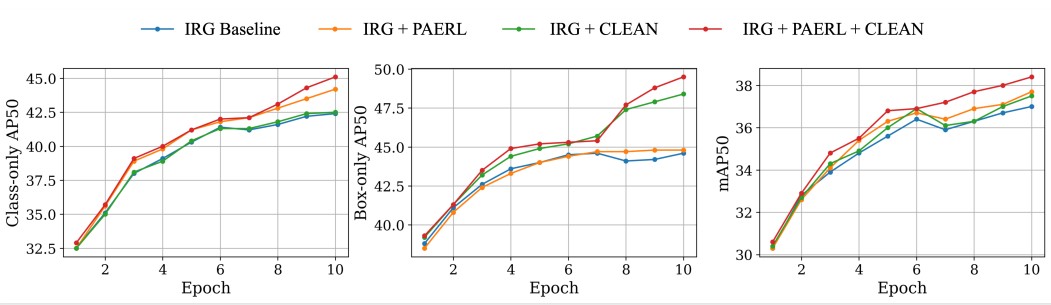

Figure 2: Teacher-model AP50 curves on the Foggy Cityscapes (Sakaridis et al., 2018) test set over training epochs - showing class-only, box-only, and full mAP. *Zoom in for best view*

Table 6: **Ablation**: Study of different components of our model.

| Method | Components | | C → F | | | | | | | | | S → C |
|--------|-----------|-------|-------|-------|-----|-------|-----|-------|--------|---------|------|-------|
| | CLEAN | PAERL | prsn | rider | car | truck | bus | train | mcycle | bicycle | mAP | AP Car |
| IRG + Ours | ✗ | ✗ | 36.9 | 45.7 | 51.5 | 26.4 | 41.3 | 25.7 | 29.1 | 39.1 | 37.0 | 45.9 |
| | ✓ | ✗ | 37.1 | 46.3 | 51.7 | 26.7 | 41.9 | 27.5 | 30.8 | 40.6 | 37.8 | 47.5 |
| | ✗ | ✓ | 37 | 46.2 | 51.7 | 27.8 | 42.7 | 28.3 | 30.8 | 40.7 | 38 | 47.4 |
| | ✓ | ✓ | 37.2 | 46.8 | 51.6 | 26.2 | 42.8 | 30.5 | 29.7 | 42.3 | **39** | **49.1** |

**How PAERL counters the long-tail class imbalance problem.** We observe that the baseline teacher produces cleaner pseudo-labels for head classes (e.g., *car*, *person*) than for tail classes (See Table 7). Quantitatively, the Pearson correlation between $\log_{10}$(class frequency) and $AP_{50}$ is $r = +0.73$ for IRG. To decouple PAERL's effect from inherent class difficulty, we correlate log-frequency with the AP gain ($\Delta$), obtaining $r_\Delta = -0.92$, indicating PAERL's bias in favor of tail classes. Indeed, the largest gains occur for rare categories such as *truck* ($+3.8$), *bus* ($+3.4$), and *train* ($+4.3$), while head classes (*car*, *person*) change by only $0.1$–$0.2$ AP. Similar trend follows for PETS and Simple-SFOD.

Table 7: Per-category $AP_{50}$ results on the target domain. We report baseline performance, performance with our method, and the improvement ($\Delta$).

| Category | Target instances | IRG | +Ours | △ IRG | PETS | +Ours | △ PETS | Simple-SFOD | +Ours | △ Simple-SFOD |
|----------|-----------------|------|-------|-------|------|-------|--------|-------------|-------|----------------|
| person | 3 419 | 36.9 | 37.0 | +0.1 | 46.1 | 46.2 | +0.1 | 40.9 | 41.0 | +0.1 |
| rider | 556 | 45.7 | 45.9 | +0.2 | 52.6 | 52.9 | +0.3 | 48.0 | 48.3 | +0.3 |
| car | 4 667 | 51.5 | 51.7 | +0.2 | 63.5 | 63.2 | −0.3 | 58.9 | 58.7 | −0.2 |
| truck | 93 | 26.4 | 30.2 | +3.8 | 21.8 | 24.0 | +2.2 | 29.6 | 33.6 | +4.0 |
| bus | 98 | 41.3 | 44.7 | +3.4 | 46.8 | 49.1 | +2.3 | 51.9 | 54.8 | +2.9 |
| train | 23 | 25.7 | 30.0 | +4.3 | 5.5 | 10.4 | +4.9 | 50.2 | 54.3 | +4.1 |
| motorcycle | 149 | 29.1 | 32.9 | +3.8 | 37.1 | 40.5 | +3.4 | 36.2 | 38.6 | +2.4 |
| bicycle | 1 175 | 39.1 | 40.6 | +1.5 | 48.7 | 48.6 | −0.1 | 44.1 | 46.2 | +2.1 |
| mAP | — | 37.0 | **39.0** | +2.0 | 40.3 | **41.9** | +1.6 | 45.0 | **46.9** | +1.9 |

**Budget Flexibility of CLEAN.** Table 12 (from Appendix A.9) demonstrates that CLEAN offers budget flexibility by accommodating both internal and external sources of foreground masks. Using simple mean-channel maps derived from the source-trained detector already yield tangible gains in mAP. Substituting stronger segmentation priors (Ren et al., 2024; Yuan et al., 2024; Lee et al., 2025) further amplifies performance. Importantly, obtaining such external priors only add a small train-time cost and zero inference overhead, as the masks are pre-computed once per target dataset before adaptation. The additional train-time cost is limited to a short, one-time preprocessing stage that is negligible compared to the target adaptation itself (see Appendix A.8 for cost details).

## 6 CONCLUSION

We identify *foreground–background confusion* as a key but previously overlooked bottleneck in Source-Free Domain Adaptive Object Detection (SFOD). To address it, we propose CLEAN, a lightweight regularizer that aligns activations with simple foreground priors, producing denoised, foreground-focused feature maps. CLEAN is budget-flexible and adds no inference overhead beyond a negligible one-time preprocessing step. Complementing this, our PAERL loss mitigates pseudo-label unreliability and dataset imbalance by emphasizing hard categories, down-weighting trivial agreements, and applying mild entropy regularization. Together, CLEAN and PAERL form a theoretically grounded, plug-and-play framework that consistently improves diverse SFOD baselines under domain shifts, charting a principled and practical path toward robust source-free detection.

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

# A    APPENDIX

In this appendix, we include the following details, which we could not include in the main paper owing to space constraints:

- Additional qualitative results [Sec. A.1]
- Proofs for the theoretical lemmas and theorems [Sec. A.2]
- Pseudo-code/Algorithm for the proposed method [Sec. A.3]
- Hyperparameter analysis [Sec. A.4]
- Datasets description [Sec. A.5]
- Disentangling Classification and Localization [Sec. A.6]
- Additional ablation studies [Sec. A.7]
- Compute and Memory analysis [Sec. A.8]
- Budget Flexibility of Clean [Sec. A.9]
- Reproducibility statement [Sec. A.10]
- Limitations and Future work [Sec. A.11]
- Declaration of LLM Usage [Sec. A.12]

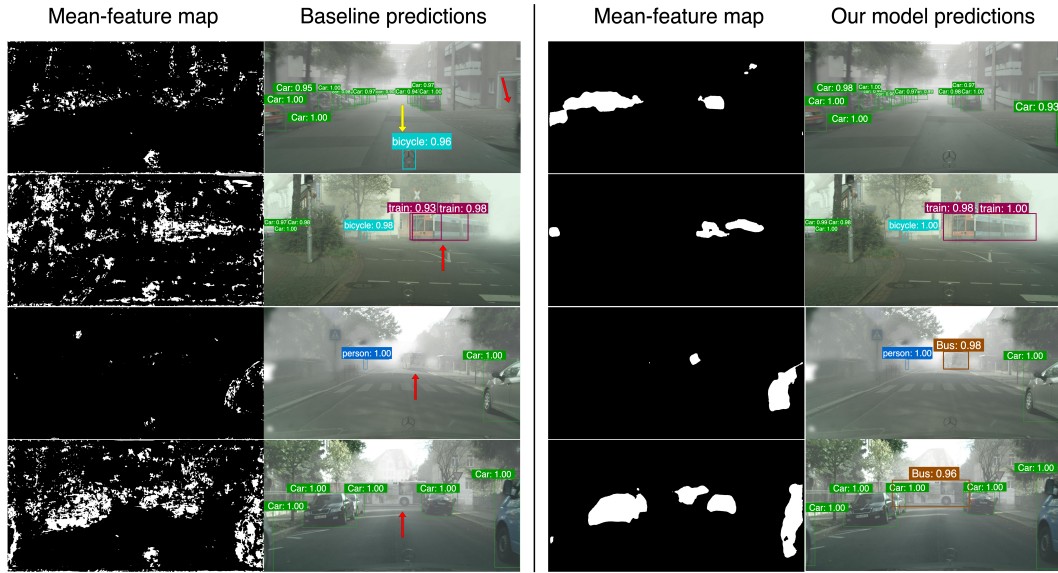

Figure 3: **Additional qualitative results.** Four examples from the Foggy Cityscapes (Sakaridis et al., 2018) target set. Mean-feature map is obtained from taking the channel-mean from the last layer of the student's backbone. Left: Baseline model - IRG (Vibashan et al., 2023) produces spurious background activations, leading to missed detections or localization errors (red arrows) and false positives (yellow arrows). Our method effectively suppresses both feature-space confusion and class-label noise, resulting in clear activations and more accurate classification and object localization. *Zoom in for best view*.

## A.1    ADDITIONAL QUALITATIVE RESULTS

Figure 3 presents additional qualitative examples comparing the baseline method (Vibashan et al., 2023) and our proposed method on the Foggy Cityscapes dataset. Each row corresponds to one scene, with the first two columns illustrating the baseline's mean-channel feature maps and predicted detections, and the last two columns showing the same representations from our method. The mean-channel map is obtained by taking the mean along the channel dimension of the last layer of the backbone, which is then upsampled to the image dimension for visualization. The baseline consistently exhibits dispersed activations, causing inaccuracies such as false positives (bicycle in the first image) and missed detections (car in the first image, incomplete train localization in the

second, and missed buses in the third and fourth images). In contrast, our method significantly reduces the irrelevant background activations, producing cleaner, focused activations that accurately highlight relevant objects and recover detections missed by the baseline. Overall, these qualitative results illustrate that our method addresses both the pseudo-label noise and foreground-background confusion that occurs due to the domain shift, resulting in improved detection accuracy and reliability.

## A.2 Proofs for the Theoretical Lemmas and Theorems

**Proof for Lemma 1**

*Proof.* For any $x$ and any true class $j \in \{0, \ldots, K\}$ define the non-negative loss vector $\ell(x) = (\ell_0(x), \ldots, \ell_K(x))^\top$ with $\ell_i(x) = -\log p_i(x) \geq 0$. Since $T_{jj} \geq \lambda$ we have

$$\ell_j(x) = \frac{T_{jj}}{T_{jj}} \ell_j(x) \leq \frac{1}{\lambda} T_{jj} \ell_j(x) \leq \frac{1}{\lambda} \sum_{i=0}^{K} T_{ji} \ell_i(x).$$

Taking expectation under the joint $(x, c)$ and using $\Pr[\hat{c} = i \mid c = j, x] = T_{ji}$ yields

$$R^{cls}_{\mathcal{D}^T,\, clean}(f^{st}_c) = \mathbb{E}_{(x,c)}\big[\ell_c(x)\big] \leq \frac{1}{\lambda} \mathbb{E}_{(x,c)}\Big[\sum_i T_{ci}\,\ell_i(x)\Big] = \frac{1}{\lambda} R^{cls}_{\mathcal{D}^T,\, noisy}(f^{st}_c),$$

which completed the proof of Lemma 1. $\qquad\square$

**Proof for Lemma 2**

*Proof.* We write the clean risk as the expectation over the two disjoint events $M = 1$ and $M = 0$:
$$\|f^{st}_r(x) - b\|_1 = M\,\|f^{st}_r(x) - b\|_1 + (1 - M)\,\|f^{st}_r(x) - b\|_1.$$

Case $M = 1$ (teacher matched the box) When $M = 1$ there exists the pseudo-box $\hat{b}$ and by the triangle inequality
$$\|f^{st}_r(x) - b\|_1 \leq \|f^{st}_r(x) - \hat{b}\|_1 + \|\hat{b} - b\|_1.$$

Case $M = 0$ (teacher missed the box) With normalised coordinates $\|f^{st}_r(x) - b\|_1 \leq 2$ for all $x, b$, hence
$$(1 - M)\,\|f^{st}_r(x) - b\|_1 \leq 2\,(1 - M).$$

Taking expectations over $\mathcal{D}^T$ and summing the two cases gives
$$R^{reg}_{\mathcal{D}^T,\, clean}(f^{st}_r) \leq \underbrace{\mathbb{E}\big[M\,\|f^{st}_r(x) - \hat{b}\|_1\big]}_{= R^{reg}_{\mathcal{D}^T,\, noisy}} + \underbrace{\mathbb{E}\big[M\,\|\hat{b} - b\|_1\big]}_{= \eta_{reg}} + 2\underbrace{\mathbb{E}[1 - M]}_{=\zeta},$$

which is exactly equation 8. $\qquad\square$

**Proof for Theorem 1**

*Proof.* Starting from the decomposition $R^{det}_{\mathcal{D}^T} = R^{cls}_{\mathcal{D}^T, clean} + R^{reg}_{\mathcal{D}^T, clean}$ (cf. Eq. 6), we apply Lemma 1 to the classification term and Lemma 2 to the regression term:
$$R^{cls}_{\mathcal{D}^T, clean}(f^{st}_c) \leq \frac{1}{\lambda} R^{cls}_{\mathcal{D}^T, noisy}(f^{st}_c), \qquad R^{reg}_{\mathcal{D}^T, clean}(f^{st}_r) \leq R^{reg}_{\mathcal{D}^T, noisy}(f^{st}_r) + \eta_{reg} + 2\,\zeta.$$
Adding the two inequalities gives equation 9. $\qquad\square$

**Proof for Theorem 2**

*Proof.* Inspired from (Wang et al., 2024), we can write,
$$R^{cls}_{\mathcal{D}^T, clean}(f^*_\eta) \leq 2\delta + \frac{2w\,\delta}{a}.$$
Combining this with the regression bound that accommodates missed boxes (Lemma 2),
$$R^{reg}_{\mathcal{D}^T, clean}(f^{st}_r) \leq R^{reg}_{\mathcal{D}^T, noisy}(f^{st}_r) + \eta_{reg} + 2\zeta,$$
and using the decomposition $R^{det} = R^{cls}_{clean} + R^{reg}_{clean}$ yields inequality equation 10. $\qquad\square$

## A.3 Psuedo-code/Algorithm of FOCUS-SFOD

---

**Algorithm 1** Training Loop of the proposed method

---

**Require:** Teacher $h^{\text{te}}$, student $h^{\text{st}}$; target images $\mathcal{X}_t$
**Require:** $(w_{\text{fg}}, w_{\text{bg}}, m, \alpha, \beta, \gamma, \lambda, \lambda_1, \lambda_2)$, optimiser $\text{Opt}(\cdot)$
1: **for each** mini-batch $\mathcal{B} \subset \mathcal{X}^t$ **do**
2:    **Augment**: obtain weak/strong views $(\tilde{x}_i^t, \bar{x}_i^t)$ for every $x_i^t \in \mathcal{B}$
3:    **Teacher forward**: $\tilde{\mathcal{Y}}_i^t \leftarrow h^{\text{te}}(\tilde{x}_i^t)$
4:    **Pseudo-labels**: $\hat{\mathcal{Y}}_i^t = \{(\hat{b}_{ij}, \hat{c}_{ij})\} \leftarrow \text{Filter}(\tilde{\mathcal{Y}}_i^t)$
5:    **Student forward**: $(\mathbf{p}_i, \mathbf{b}_i) \leftarrow h^{\text{st}}(\bar{x}_i^t)$
6:    Apply Eq. 3 to $\mathbf{p}_i$ to obtain $\mathbf{p}_i'$
7:    $w_{\hat{c}} \leftarrow \begin{cases} w_{\text{fg}}, & \hat{c} \text{ is foreground,} \\ w_{\text{bg}}, & \text{otherwise} \end{cases}$
8:    Compute $\mathcal{L}_{\text{PAERL}}$ via Eq. 4
9:    **Compute masks**: obtain $A_{\text{G}}$ (from source-trained model or external segmentation model)
             compute student mean maps $A_{\text{S}}(x_i^t) = \text{mean-channel}\big(g^{\text{st}}(x_i^t)\big)$
10:   Compute $\mathcal{L}_{\text{CLEAN}}$ via Eq. 5
11:   Compute standard detection loss $\mathcal{L}_{reg}$
12:   Aggregate and update: $\theta^{\text{S}} \leftarrow \text{Opt}\big(\theta^{\text{st}}, \nabla_{\theta^{st}}(\mathcal{L}_{\text{PAERL}} + \mathcal{L}_{\text{CLEAR}} + \mathcal{L}_{reg})\big)$
13: **Update teacher**: $\theta^{te} \leftarrow \lambda\theta^{te} + (1 - \lambda)\theta^{st}$
14: **return** Adapted teacher $h^{\text{te}}$

---

## A.4 Hyperparameter Analysis

The eight hyperparameters shape how our method balances robustness against the pseudo class-label noise and the foreground-background confusion. $w_{\text{fg}}$ and $w_{\text{bg}}$ are class weights that scale every foreground and background box in PAERL, preventing the large number of background pseudo-boxes from overwhelming the learning process. The parameter $m$ controls the peak-squeezing transform in Eq. 4: a larger $m$ makes the predictions closely approximate a one-hot encoding so that one mislabeled box cannot dominate the gradient. The triplet $(\alpha, \beta, \gamma)$ balances PAERL's three sub-terms: $\alpha$ sets the strength of the peak-adjusted cross-entropy (driving class discrimination), $\beta$ weights the MAE term that encourages calibration and complements CE's unbounded gradient offering robustness against the noisy pseudo-class labels, and $\gamma$ governs the global KL regularizer that keeps the per-batch class distribution close to uniform, reducing confirmation bias. Finally, $\lambda_1$ and $\lambda_2$ weight CLEAN's mean-$\ell_1$ alignment and Dice consistency, respectively; $\lambda_1$ tightens pixel-wise correspondence between student activations and the binary masks, while $\lambda_2$ focuses on region-level overlap, making CLEAN robust when the external mask slightly over or under segments. Together, these parameters let PAERL curb pseudo class-label noise, let CLEAN correct spatial drift, and still leave room for the localization loss to fine-tune boxes. Table 8 demonstrates the hyperparameter analysis results. As we can see, our method is not overly sensitive to the choice of hyperparameters indicating its robustness. We use Ren et al. (2024) in CLEAN for the hyperparameter analysis.

## A.5 Dataset Descriptions

We use five publicly available datasets covering four domain-shift scenarios. Cityscapes (Cordts et al., 2016) is an urban street scene dataset comprising 5,000 finely annotated images collected from diverse cities and seasons, from which we use 2,925 images for training and 500 for validation. It includes eight categories: person, rider, car, truck, bus, train, motorcycle, and bicycle. Foggy Cityscapes (Sakaridis et al., 2018) extends Cityscapes by overlaying synthetic fog at three intensity levels (0.005, 0.01, and 0.02) to simulate poor visibility conditions. KITTI (Geiger et al., 2013) is a well-known autonomous driving dataset consisting of 7,481 real-world street scene training images. Sim10k (Johnson-Roberson et al., 2016) provides 10,000 synthetic urban scene images of cars, rendered from the video game Grand Theft Auto. BDD100k (Yu et al., 2018) is a large-scale dataset comprising 100,000 driving scene images captured across various weather conditions and times of day. To demonstrate the efficacy of our method on extreme domain shifts, we use following

Table 8: Hyperparameter analysis for *Cityscapes→ Foggy Cityscapes* on IRG. ■ marks the hyperparameter being varied.

| $w_{\text{fg}}$ | $w_{\text{bg}}$ | $m$ | $\alpha$ | $\beta$ | $\gamma$ | $\lambda_1$ | $\lambda_2$ | mAP |
|---|---|---|---|---|---|---|---|---|
| 1 | | | | | | | | 37.8 |
| 2 | 1 | 1e4 | 0.5 | 0.1 | 0.01 | 2 | 4 | **38.4** |
| 3 | | | | | | | | 38.2 |
| 4 | | | | | | | | 38.2 |
| | 1 | | | | | | | **38.4** |
| 2 | 2 | 1e4 | 0.5 | 0.1 | 0.01 | 2 | 4 | 37.7 |
| | 3 | | | | | | | 37.3 |
| | 4 | | | | | | | 36.8 |
| | | 200 | | | | | | 37.1 |
| 2 | 1 | 500 | 0.5 | 0.1 | 0.01 | 2 | 4 | 37.3 |
| | | 1e3 | | | | | | 38.3 |
| | | 1e4 | | | | | | **38.4** |
| | | | 0.1 | | | | | 37.4 |
| 2 | 1 | 1e4 | 0.5 | 0.1 | 0.01 | 2 | 4 | **38.4** |
| | | | 1 | | | | | 37.5 |
| | | | 2 | | | | | 37.8 |
| | | | | 0.1 | | | | **38.4** |
| 2 | 1 | 1e4 | 0.5 | 0.5 | 0.01 | 2 | 4 | 38 |
| | | | | 1 | | | | 37.6 |
| | | | | 2 | | | | 37 |
| | | | | | 0.01 | | | **38.4** |
| 2 | 1 | 1e4 | 0.5 | 0.1 | 0.05 | 2 | 4 | 38.1 |
| | | | | | 0.1 | | | 37.6 |
| | | | | | 0.5 | | | 37.5 |
| | | | | | | 1 | | 37.7 |
| 2 | 1 | 1e4 | 0.5 | 0.1 | 0.01 | 2 | 4 | **38.4** |
| | | | | | | 3 | | 38.1 |
| | | | | | | 4 | | 37.8 |
| | | | | | | | 1 | 37.5 |
| 2 | 1 | 1e4 | 0.5 | 0.1 | 0.01 | 2 | 2 | 37.7 |
| | | | | | | | 3 | 37.9 |
| | | | | | | | 4 | **38.4** |

datasets: PascalVOC (Everingham et al., 2010), COCO (Lin et al., 2014), FLIR (Teledyne FLIR LLC, 2019), and Clipart (Inoue et al., 2018).

## A.6   DISENTANGLING CLASSIFICATION AND LOCALIZATION.

To clearly distinguish between improvements in classification quality and localization quality, we introduce two variants of the standard VOC AP50: Class-only AP50 and Box-only AP50. These metrics use the same detection outputs as the standard evaluation; only the scoring process is altered, adding zero inference cost.

i) For Class-only AP50, each predicted bounding box is snapped to its nearest ground-truth box if their IoU is $\geq 0.5$. After this, the predicted class labels remain unchanged. As the boxes now have perfect localization, the metric exclusively evaluates the correctness of predicted labels. Thus, improvements in Class-only AP directly measure reduced class-label noise, validating classification accuracy. Our experiments show PAERL improving this metric by approximately 2-3 AP points over the baseline (Fig. 2 left), validating its effectiveness in correcting noisy class labels. ii) In contrast, Box-only AP50 removes class information by collapsing all predicted and ground-truth labels into a single generic object class. This metric assesses only whether predicted boxes overlap sufficiently (IoU $\geq 0.5$) with ground-truth boxes, thereby isolating spatial localization performance. CLEAN raises this metric by approximately 3-4 AP points compared to the baseline (Fig. 2 middle), demonstrating its strength in improving spatial alignment of the predicted boxes. By combining PAERL and CLEAN, the detector simultaneously benefits from reduced class-label noise and improved localization, resulting in enhanced overall detection performance (Fig. 2 right).

## A.7   ADDITIONAL ABLATION ON PAERL COMPONENTS

Table 9 presents the ablation analysis of Foreground-Background weighting (FG-BG) & Entropy components. Removing FG–BG weighting and entropy regularization (Row 3, $\varepsilon$-softmax) causes a clear drop in mAP below the IRG baseline, showing their necessity. While adding CLEAR alone (Row 4) recovers some performance, the best results are obtained when both CLEAR and PAERL's two terms are jointly applied (Row 2). This confirms that off-the-shelf $\varepsilon$-softmax alone is insufficient for robust SFOD adaptation.

Table 9: **Ablation**: Study on different components of PAERL.

| Method | CLEAR | FG-BG & Entropy | mAP |
|---|---|---|---|
| IRG (rep) | — | — | 37.0 |
| IRG + Ours (PAERL full) | ✓ | ✓ | 39.0 |
| $\varepsilon$-softmax (Wang et al., 2024) | ✗ | ✗ | 36.7 |
| $\varepsilon$-softmax (Wang et al., 2024) + CLEAR | ✓ | ✗ | 37.6 |

## A.8   COMPUTATION COST AND MEMORY ANALYSIS

Tables 10 and 11 report the additional cost of integrating GSAM (Ren et al., 2024) into our pipeline. The time overhead is marginal: the offline GSAM pass completes in $1\,050\,\text{s}$ ($\sim 17\,\text{min}$), which is only $3.8\%$ of the $28\,000\,\text{s}$ required for our model training (and only $5.2\%$ relative to the baseline). When added to the full pipeline, the end-to-end wall-clock increases only slightly ($28\,084\,\text{s} \rightarrow 29\,134\,\text{s}$), well within the typical run-to-run variability of large-scale training. The memory overhead is also short-lived: GSAM peaks at $18\,\text{GB}$ only during its 17-minute preprocessing, while training itself never exceeds $9.6\,\text{GB}$. Since these stages do not overlap, the entire procedure fits comfortably on a single $24\text{-}48\,\text{GB}$ GPU without any modification to training. Moreover, the amortization cost is small: GSAM masks are generated once per target split and can be cached for reuse in all subsequent experiments. The extraction step is fully parallel across images, so on a multi-GPU node the elapsed time approaches standard data-loading latency. For completeness, we note that ESC-Net (Lee et al., 2025) and OV-SAM (Yuan et al., 2024) are both lighter than Grounded-SAM in parameter size, and therefore require less compute and memory; we report GSAM values here since it represents the most demanding case among the three. Overall, even with GSAM enabled, the complete adaptation run finishes in under 8 hours and $< 18\,\text{GB}$ peak memory on a single RTX A6000, confirming that the footprint remains modest.

Table 10: Computation time comparison.

| Setting | Offline GSAM time (1000s) | Training time (1000s) | Test time (s) | End-to-end time (1000s) |
|---|---|---|---|---|
| IRG (baseline) | – | 20 | 84 | 20.08 |
| IRG + Our target adaptation | – | 28 | 84 | 28.08 |
| IRG + Ours + GSAM masks | 1.050 | 28 | 84 | 29.13 |

Table 11: Memory usage comparison.

| Setting | Offline GSAM peak mem (GB) | Training peak mem (GB) | Stage-wise peak mem (GB) |
|---|---|---|---|
| IRG (baseline) | – | 6.9 | 6.9 |
| IRG + Our target adaptation | – | 9.6 | 9.6 |
| IRG + Ours + GSAM masks | 18.4 | 9.6 | 18.4 |

## A.9 BUDGET FLEXIBILITY OF CLEAN

Table 12 reports results for CLEAN using different mask sources, demonstrating that our approach delivers consistent gains while offering budget flexibility. The external binary masks are generated by simply thresholding foreground versus background classes. Importantly, we do not exploit any label information from these models, thereby avoiding knowledge leakage. Instead, the masks serve only as structural priors, guiding the student model to better distinguish between foreground and background regions.

Table 12: Performance on 'Cityscapes→Foggy Cityscapes' and 'Sim10k→Cityscapes' with different binary masks in CLEAN.

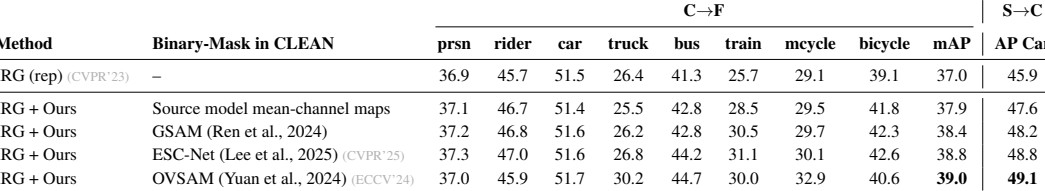

| Method | Binary-Mask in CLEAN | C→F | | | | | | | | | S→C |
|---|---|---|---|---|---|---|---|---|---|---|---|
| | | prsn | rider | car | truck | bus | train | mcycle | bicycle | mAP | AP Car |
| IRG (rep) (CVPR'23) | – | 36.9 | 45.7 | 51.5 | 26.4 | 41.3 | 25.7 | 29.1 | 39.1 | 37.0 | 45.9 |
| IRG + Ours | Source model mean-channel maps | 37.1 | 46.7 | 51.4 | 25.5 | 42.8 | 28.5 | 29.5 | 41.8 | 37.9 | 47.6 |
| IRG + Ours | GSAM (Ren et al., 2024) | 37.2 | 46.8 | 51.6 | 26.2 | 42.8 | 30.5 | 29.7 | 42.3 | 38.4 | 48.2 |
| IRG + Ours | ESC-Net (Lee et al., 2025) (CVPR'25) | 37.3 | 47.0 | 51.6 | 26.8 | 44.2 | 31.1 | 30.1 | 42.6 | 38.8 | 48.8 |
| IRG + Ours | OVSAM (Yuan et al., 2024) (ECCV'24) | 37.0 | 45.9 | 51.7 | 30.2 | 44.7 | 30.0 | 32.9 | 40.6 | **39.0** | **49.1** |

## A.10 REPRODUCIBILITY STATEMENT

We release the complete anonymized code for our method as a supplementary zip file. Implementation details necessary for reproduction are provided in Sec. 5. As a plug-and-play approach, our method does not alter any training configurations of the baseline models. The best hyperparameters specific to our method are reported in Sec. A.4.

## A.11 LIMITATIONS AND FUTURE WORK

While the current work focuses on SFOD methods built on Faster-RCNN, future work includes extending our approach to transformer-based and anchor-free detectors, and exploring fully mask-free regularization strategies to further simplify adaptation.

## A.12 DECLARATION OF LARGE LANGUAGE MODELS USAGE:

We used large language models (LLMs) solely for polishing our writing and performing grammar checks. No part of the technical content, analysis, or conclusions was generated by LLMs.

