# OpenReview forum: "Foreground Confusion under Domain Shift: The Hidden Bottleneck in Source‑Free Domain Adaptive Object Detection"
_ICLR.cc/2026/Conference — ICLR 2026 Conference Withdrawn Submission_

### Official Review · Reviewer_1NCk · 2025-10-26

**Soundness:** 3
**Presentation:** 3
**Contribution:** 3
**Rating:** 4
**Confidence:** 5

**Summary:**

This paper addresses a critical challenge in Source-Free Domain Adaptive Object Detection (SFOD), a task where detectors are adapted to unlabeled target domains without access to source data (critical for privacy/storage-constrained scenarios). The work identifies two key bottlenecks in SFOD: (1) unreliable pseudo-labels and (2) foreground-background confusion—a previously overlooked issue where domain shift causes spurious background activations, degrading localization and cascading into misclassification. To mitigate these, the authors propose FOCUS-SFOD, a lightweight, architecture-agnostic framework with two complementary components, CLEAN and PAERL. The paper provides theoretical risk-bound analyses linking CLEAN and PAERL to tighter localization and classification errors.

**Strengths:**

1. This paper is well-written and easy to follow.
2. This paper focuses on an important problem, SFOD. In real applications, the target data may be unavailable due to privacy issues.
3. Evaluations span diverse domain shifts (adverse weather, synthetic to real, cross to camera, extreme shifts like RGB to Thermal) and five public datasets, ensuring generalizability.

**Weaknesses:**

1. As stated in the paper, the FOCUS-SFOD is an architecture-agnostic framework. The method is only evaluated on Faster R-CNN. It is unclear how FOCUS-SFOD would perform on modern detectors like DETR, anchor-free models (e.g., FCOS).
2. Vague Foreground Prior Details. While CLEAN supports multiple mask sources, the paper provides limited guidance on when to choose which prior. How do internal vs. external priors perform under extreme domain shifts (e.g., Thermal to RGB)?
3. While the paper claims to "reproduce each baseline following its published code and hyper-parameters," it does not provide side-by-side comparisons of reproduced vs. original baseline results (e.g., how close is the reproduced IRG mAP to the original CVPR’23 paper?).

**Questions:**

1. What is the difference between the proposed PAERL (Peak-Adjusted Entropy-Regularized Loss) and the epsilon-softmax?

---

### Official Review · Reviewer_ZPpd · 2025-10-28

**Soundness:** 3
**Presentation:** 3
**Contribution:** 2
**Rating:** 4
**Confidence:** 4

**Summary:**

This paper studies source-free domain adaptation for object detection (SFOD) and identifies two major challenges in this setting: (1) unreliable pseudo-labels and (2) foreground–background confusion. To address these issues, the authors propose FOCUS-SFOD, which introduces two components: a Consistency Loss for Eliminating Activation Noise and a Peak-Adjusted Entropy-Regularized Loss. Experiments conducted on multiple widely used benchmarks demonstrate the effectiveness of the proposed approach.

**Strengths:**

The paper provides a clear identification and analysis of two important problems in SFOD. The proposed method is well-motivated, and the theoretical reasoning behind the two designed loss functions is clearly explained. The ablation studies and analysis experiments are comprehensive, which strengthens the empirical claims.

**Weaknesses:**

While the paper identifies foreground–background confusion as the main bottleneck of SFOD, the experiments are based solely on the Faster R-CNN framework with a ResNet backbone. These architectures are somewhat outdated in the current object detection literature. It remains unclear whether the same issue persists when using more recent detectors, such as DETR-based models, ViT-based detectors, or even vision–language models (VLMs), which have shown much stronger feature alignment capabilities. The authors should discuss whether the foreground–background confusion is inherently tied to the SFOD setting, or mainly a result of using weaker detectors and backbones. Additional experiments on more recent architectures would provide stronger evidence for the generality of the claimed problem and the proposed method.

Moreover, the proposed two losses, i.e., a regularized cross-entropy loss for stabilizing training and a consistency loss, are conceptually similar to techniques already common in unsupervised domain adaptation (UDA), especially in mean teacher-style frameworks (e.g., [a,b]). The paper should more explicitly clarify the conceptual difference and novelty of their approach compared to these prior works. It would also strengthen the paper if the authors could include comparative experiments or ablations to demonstrate the advantages of their formulation beyond existing consistency-based or entropy-regularized UDA methods.

[a] Prototypical Pseudo Label Denoising and Target Structure Learning for Domain Adaptive Semantic Segmentation

[b] COMET: Contrastive Mean Teacher for Online Source-Free Universal Domain Adaptation

**Questions:**

Please see the Weakness.

---

### Official Review · Reviewer_atja · 2025-10-31

**Soundness:** 3
**Presentation:** 2
**Contribution:** 3
**Rating:** 4
**Confidence:** 4

**Summary:**

The paper introduces two losses meant to improve performance of source-free domain adaptive object detection. The first loss (CLEAN) forces the student’s final feature maps to agree with prior foreground segmentations obtained using SAM-based methods. The second loss (PAERL) reweights traditional cross-entropy towards hard cases. The evaluation shows slight improvements over previous state-of-the-art source-free domain adaptive object detection methods.

**Strengths:**

The method is complementary to existing techniques in source-free domain adaptive object detection and yields state-of-the-art results on common benchmarks. The PAERL and CLEAN loss components make intuitive sense and are well motivated. Figure 2 nicely illustrates the contributions of each of the losses.

**Weaknesses:**

While prior work in source-free domain-adaptive object detection is well covered, I feel like there are missing connections to general object detection literature. For example PAERL seems to be conceptually similar to focal loss, yet focal loss similar methods to focus on hard cases are not discussed. Furthermore, the proposed CLEAN loss component seems somewhat unprincipled, forcing pixel-wise feature map means towards 0 or 1 to minimize disagreement with a binary prior segmentation (see questions).

**Questions:**

* Figure 1:
  * The “channel-mean feature maps” seem like binary segmentations to me. Is this because they are encouraged by CLEAN to effectively be binary? If so, the figure caption is confusing when talking about brighter regions corresponding to higher activations.
* CLEAN
  * Why use both an l1-loss and a dice loss? Shouldn’t one of the two suffice?
  * Why use a binary prior foreground-background segmentation and not a continuous map of foreground-background probabilities that is able to better capture ambiguities within the data?
  * The loss seems to encourage the pixel-wise mean of the final feature maps to be either 0 or 1, which seems somewhat inelegant and like it could waste model capacity to achieve. It seems like this could be alleviated by either adaptively choosing a continuous threshold (in the binary case) or treating the mean feature map as foreground logit (in the continuous case).
  * Intuitively, why is CLEAN superior to just filtering pseudolabels using the prior segmentation?
* “domain shift causes the feature space to become entangled between foreground and background regions” \- what does entangle mean here exactly? I feel like this statement should either be rephrased or backed up with evidence.

---

### Official Review · Reviewer_uskL · 2025-10-31

**Soundness:** 3
**Presentation:** 3
**Contribution:** 2
**Rating:** 6
**Confidence:** 4

**Summary:**

This paper identifies foreground-background confusion (spurious background activations caused by domain shift) as a key bottleneck in Source-Free Domain Adaptive Object Detection (SFOD), alongside unreliable pseudo-labels. The authors introduce FOCUS-SFOD, a lightweight framework with two complementary losses: CLEAN mitigates this confusion by aligning activation maps with foreground priors to improve localization, while PAERL reduces sensitivity to noisy pseudo-labels. This theoretically-grounded, architecture-agnostic method delivers consistent gains without inference overhead.

**Strengths:**

1. The work formally identify and analyze foreground-background confusion as a fundamental bottleneck in SFOD, showing it's a primary cause of localization and classification degradation.

2. The method achieves consistent and significant gains across multiple strong baselines (IRG, PETS, Simple-SFOD) and diverse domain shifts, improving performance by up to +3.9 mAP.

3. The PAERL loss is shown to effectively combat class imbalance by encouraging learning on harder or underrepresented categories. Experiments confirm it provides the largest AP gains for rare categories.

**Weaknesses:**

1. Reliance on Strong External Priors: The primary results (Tables 1-5) are generated using foreground priors from powerful, general-purpose segmentation models like OVSAM. These segmentation models are significantly stronger and trained on more diverse data than the Faster R-CNN detector being adapted. This introduces a potential confound: it is unclear how much of the performance gain comes from the proposed CLEAN loss versus the implicit knowledge distillation from these superior external models. While the pape includes an ablation study in Appendix A.9 (Table 12) that explores different priors and shows gains even with a weaker "source model" prior, the main claims of the paper rely on this "strong-teacher" setup.

2. Increased Hyperparameter Complexity: The proposed framework introduces eight new hyperparameters ($w_{fg}$, $w_{bg}$, $m$, $\alpha$, $\beta$, $\gamma$, $\lambda_{1}$, $\lambda_{2}$) that must be tuned to balance the PAERL and CLEAN losses. This represents a significant increase in tuning complexity compared to the baseline methods. Although the authors provide a robustness analysis in Table 8, finding the optimal combination of these parameters for a new dataset could be a practical challenge.

3. Additional Preprocessing Cost: While the framework has zero inference overhead, the CLEAN loss requires a non-trivial, one-time preprocessing step to compute and cache the binary foreground masks for the entire target dataset before adaptation can begin. Although this cost is amortized, it is an additional computational hurdle not required by the baselines.

**Questions:**

Please see the Weaknesses above.

---

### Note · Authors · 2025-11-14

I have read and agree with the venue's withdrawal policy on behalf of myself and my co-authors.